# Advanced Bronchoscopic Technologies for Biopsy of the Pulmonary Nodule: A 2021 Review

**DOI:** 10.3390/diagnostics11122304

**Published:** 2021-12-08

**Authors:** Micah Z. Levine, Sam Goodman, Robert J. Lentz, Fabien Maldonado, Otis B. Rickman, James Katsis

**Affiliations:** 1Rush University Medical Center, Department of Internal Medicine, Division of Pulmonary and Critical Care, Rush University, Chicago, IL 60612, USA; micah_z_levine@rush.edu (M.Z.L.); samuel_n_goodman@rush.edu (S.G.); 2Vanderbilt University Medical Center, Department of Internal Medicine, Division of Pulmonary and Critical Care, Vanderbilt University, Nashville, TN 37232, USA; robert.j.lentz@vanderbilt.edu (R.J.L.); fabien.maldonado@vanderbilt.edu (F.M.); otis.rickman@vanderbilt.edu (O.B.R.); 3Rush University Medical Center, Department of Surgery, Division of Cardiothoracic Surgery, Rush University, Chicago, IL 60612, USA

**Keywords:** bronchoscopy, lung cancer, lung nodule, navigational bronchoscopy, robotic bronchoscopy

## Abstract

The field of interventional pulmonology (IP) has grown from a fringe subspecialty utilized in only a few centers worldwide to a standard component in advanced medical centers. IP is increasingly recognized for its value in patient care and its ability to deliver minimally invasive and cost-effective diagnostics and treatments. This article will provide an in-depth review of advanced bronchoscopic technologies used by IP physicians focusing on pulmonary nodules. While most pulmonary nodules are benign, malignant nodules represent the earliest detectable manifestation of lung cancer. Lung cancer is the second most common and the deadliest cancer worldwide. Differentiating benign from malignant nodules is clinically challenging as these entities are often indistinguishable radiographically. Tissue biopsy is often required to discriminate benign from malignant nodule etiologies. A safe and accurate means of definitively differentiating benign from malignant nodules would be highly valuable for patients, and the medical system at large. This would translate into a greater number of early-stage cancer detections while reducing the burden of surgical resections for benign disease. There is little high-grade evidence to guide clinicians on optimal lung nodule tissue sampling modalities. The number of novel technologies available for this purpose has rapidly expanded over the last decade, making it difficult for clinicians to assess their efficacy. Unfortunately, there is a wide variety of methods used to determine the accuracy of these technologies, making comparisons across studies impossible. This paper will provide an in-depth review of available data regarding advanced bronchoscopic technologies.

## 1. Introduction

Interventional pulmonology (IP) has come a long way since its introduction by Gustav Killian when he removed a pork bone from a farmer’s lung in 1876 [1]. It was not until nearly 100 years later, in 1972, that Howard Anderson attempted the first tissue sampling of lung parenchyma through a rigid bronchoscope [1]. Bronchoscopy has evolved dramatically since and today includes embodied robotics, advanced navigational systems, and intraoperative imaging technologies including cone beam CT (computerized tomography), digital tomosynthesis, and endobronchial ultrasonography. The accelerating rate of technological advancement is challenging to keep up with for clinicians both in and outside of the field of IP. This review aims to synthesize the literature around these technologies so that clinicians can gain insight into what these technologies could offer them, their patients, and their practices. While reviewing the literature, we quickly uncovered that reporting of diagnostic yield, benign nodule follow-up and benign histology was highly variable. The lack of consensus definitions makes comparison across studies impossible. We present below the most up-to-date review of the literature; however, no ”best” technology can be determined without additional study and consensus agreement on how diagnostic yield is determined. Given the very high incidence of pulmonary nodules and the significant consequences of missed or delayed diagnoses, research in this field is of great importance and consequence to our patients.

Lung cancer is the most fatal and the second most common cancer worldwide [2]. A lung nodule can represent the earliest detectable stage of lung cancer. It has been well demonstrated that the stage of diagnosis is inversely related to prognosis, with early detection leading to significant improvements in survival. Five-year survival for stage 1a lung cancer after resection is estimated to be 90%, while stage 4 cancer has a five-year survival of around 10% [2].

Recently, two large randomized controlled trials (RCT) demonstrated improved survival with lung cancer screening of high-risk individuals. The national lung cancer screening trial (NLCST) compared the survival for the patient undergoing yearly low dose CT scans versus those who had annual chest X-rays [3]. Annual Screening with low dose CT (LDCT) of the chest demonstrated a survival benefit and one of the lowest numbers needed to screen (NNS) among routine cancer screening modalities. The NNS for lung cancer screening is 320 to prevent one death [3], while the NNS with mammography for breast cancer is estimated around 781 and colonoscopy for colorectal cancer is 1250 [4]. The benefit of lung cancer screening was again demonstrated in the Nelson Trial, but with a lower risk population [5]. This new information led to the expansion of screening criteria supported by the United States Preventive Task Force (USPSTF) to include patients with a 20 pack-year smoking history and 50 years of age compared to 30 pack-years and 55 years of age as studied in the NLST [6].

Despite robust and reproduced evidence of the benefit of LDCT lung cancer screening, only 5% of eligible patients undergo LDCT screening nationwide [6]. Even with this low screen rate, the annual incidence of lung nodules is estimated to be 1.5 million [7]. The incidence is anticipated to rise as more patients are screened, as is the frequency with which chest CT scans are being performed [8].

As the number of patients with lung nodules increases, there will be increased demand to perform tissue sampling. Guidelines advocate for calculation of pre-test probability of malignancy to help determine which nodules should be subjected to an attempted biopsy [9]. This may be performed by expert lung nodule clinicians or validated prediction models or calculators, which typically take radiographic and demographic characteristics into account. However, there are no large prospective randomized controlled trials to guide clinicians on the best practices for tissue sampling.

Traditionally, there are three options for tissue sampling of the lung nodule: surgical resection (video-assisted thorascopic or open), CT-guided transthoracic needle biopsy (CT-TTNB), or bronchoscopic biopsy. Choosing a biopsy modality requires careful consideration of finding the least invasive option with the highest diagnostic yield. Additionally, as tumor specimens are increasingly being sent for advanced molecular analysis, the volume of tissue recovered by a biopsy technique is an emerging consideration in biopsy modality as well, particularly when there is clinical suspicion of more advanced stage lung cancer [10].

Surgical resection is nearly always diagnostic and has the advantage of being simultaneously therapeutic in limited stage disease. However, it is the most invasive modality, with more risk compared to the minimally invasive diagnostic alternatives, and operations performed for nodules which are ultimately determined to be benign are considered futile thoracotomies. CT-TTNB, through meta-analyses, has been shown to have excellent diagnostic yield up to 92% [11], but carries with it a high risk of pneumothorax (20%) and rates of bleeding complications [12]. Furthermore, rates of successful molecular analysis on TTNB specimens are relatively low, at 31.8%, 27.3%, and 35.3% for EGFR, KRAS, and ALK, respectively [13]. The high incidence of complications may cause understandable hesitancy for clinicians in referral, potentially leading to delays in diagnosis. Another downside to this diagnostic modality is the inability to stage the mediastinum simultaneously.

Traditional bronchoscopic biopsy, guided only by fluoroscopy, has historically had a low diagnostic yield, with diagnostic rates for nodules under 2 cm estimated to be 34% and still only 63% for lesions over 3 cm [14]. Bronchoscopy has a notably better safety profile compared to CT-TTNB, with pneumothorax complicating only 1.5% and bleeding occurring in fewer than 1% of cases [15]. More recent series involving contemporary advanced navigational platforms, however, report diagnostic yields approaching those typically reported in series of CT-TTNB, while maintaining the same safety profile. Additionally, mediastinal staging can be performed during the same bronchoscopic procedure, obviating the need for a second intervention.

In this review, we will lay out existing advanced bronchoscopic techniques and technologies for the biopsy of pulmonary nodules with a particular focus on available clinical data related to yield and complication rates. The advanced bronchoscopic technologies reviewed herein include non-guided bronchoscopy, thin/ultrathin bronchoscopes, radial probe endobronchial ultrasound (REBUS), virtual bronchoscopic navigation (VBN), electromagnetic navigational bronchoscopy (ENB), cone-beam CT (CBCT) assisted bronchoscopy, digital tomosynthesis assisted bronchoscopy and robotic bronchoscopy.

## 2. Non-Guided Bronchoscopy

To begin our discussion on advanced bronchoscopy, we must start with the basics of non-guided bronchoscopy as a foundation. Central lesions are far easier to biopsy and allow for superior diagnostic yield with traditional bronchoscopic techniques compared to the sampling of peripherally located nodules. Transbronchial biopsy (TBx) of these large, centrally located nodules have yields approaching 74%, whereas sampling of smaller (<2 cm) lesions in the peripheral third of the chest have yields as low as 14%. Overall diagnostic yield is estimated at 57% for all lesions and 34% for lesions < 2 cm [16]. In a single-center retrospective analysis of 207 patients, the overall sensitivity of non-guided bronchoscopy was 25.6%, and sensitivities for bronchial aspiration, bronchoalveolar lavage (BAL), and brushing were 14.2%, 11.6% and 16.5%, respectively [17]. Factors that increased yield included larger lesion, central location, presence of bronchus sign and higher standardized uptake value (SUV) on PET. The bronchus sign refers to a nodule to which an airway directly leads, as seen on CT [18]. Even with optimal factors (tumor >4 cm, central location, bronchus sign and SUV > 12), the overall sensitivity for bronchoscopy was around 40% [17]. This is both surprising and discouraging, given that the pooled sensitivity of sputum cytology for central endobronchial lesions is estimated to be 88% [14].

A metanalysis of 18 different studies for non-guided transbronchial needle aspiration (TBNA) of peripheral nodules found a pooled diagnostic yield of 53%, with yield increased to 70% when CT bronchus sign was present. When rapid onsite cytological evaluation (ROSE) was used, the pooled yield was 62% versus 51% when absent. Malignant nodules had a more successful yield at 55% when compared to benign at 17% [19]. Non-guided bronchoscopy offers acceptable performance in only selected situations, including larger lesions, central location, presence of CT bronchus sign, high pre-test probability for malignancy, and availability of ROSE. If these metrics are not met, then traditional non-guided bronchoscopic biopsy may offer unacceptable yield, necessitating consideration of more advanced technologies.

A 2012 meta-analysis of over 3000 nodules biopsied with guided bronchoscopy demonstrated a pooled diagnostic yield of 70%, a yield far superior to traditional bronchoscopic biopsy. The studies included in the meta-analysis utilized VB, REBUS and EMN (electromagnetic navigation), guide sheath (GS), and ultrathin bronchoscopy. The yield increased with the size of the lesion, and complication rates remained low at 1.6% pneumothorax and 0.7% major bleeding [20]. Other studies have demonstrated higher yields even up to 94% at a single center in New York [21]. Below, we will outline these and other advanced bronchoscopic technologies in detail.

## 3. Thin and Ultra-Thin Bronchoscopy

One major limitation of conventional bronchoscopy lies with the anatomic constraints of the physical bronchoscope and its inability to reach distal subsegmental levels owing to the bronchoscope’s large outer diameter and restricted range of motion. While there is no official standard bronchoscope size, the typical outer diameter (OD) is 5–6 mm [22] which can reach the third and fourth generation bronchi at best. Due to its smaller size and flexibility, thin and ultrathin bronchoscopes (OD < 3 mm) can traverse further into the lung periphery, often reaching the ninth bronchial generation, thus gaining improved access to peripheral lesions for tissue sampling [23].

Thin/ultrathin bronchoscopy is often combined with other guided techniques, such as CT guidance, Virtual Bronchoscopic Navigation (VBN), and Radial Probe Endrobronchial Ultrasound (REBUS), to improve lesion localization. There are no head-to-head studies comparing thin bronchoscopy (TB) alone versus standard bronchoscopy. A 2004 single-center study found a sensitivity of 65% when using an ultrathin bronchoscope (UTB) with the aid of VBN for CT guidance [24]. Another prospective study included 71 patients undergoing EBUS-TBB using a 3.4 mm thin scope and had a yield of 82% for lesions >2 cm and 67% for those <2 cm [25].

One retrospective study comprising 44 of 338 patients who underwent bronchoscopy evaluated whether substituting a TB with the UTB during multimodal bronchoscopy improved lesion ultrasound visualization and diagnostic yield (DY). After substitution, in cases where the radial probe was within the target lesion (a concentric view), the diagnostic yield was 80%. The yield decreased to 72% when the probe is adjacent to the lesion (eccentric) and 0% with no visible lesion. Overall diagnostic yield was 65% [26]. This demonstrated that substitution of TB for UTB as needed improved position of REBUS probe, at times converting an eccentric view to a concentric view and from no view to an eccentric view (Figure 1). With an improvement in view, there was an increase in diagnostic yield.

A 2015 trial from Japan randomized 310 patients who underwent transbronchial biopsy with REBUS, fluoroscopy and VBN to either ultrathin bronchoscope (3 mm) or thin bronchoscope (4 mm) with guide sheath. The ultrathin scope could reach more distal bronchi (median 5th vs. 4th generation) and had a higher diagnostic yield of 74% compared to 59%. Complications occurred in 3% vs. 5%, respectively [27].

In a larger, more recent trial from 2019, patients were randomized to undergo EBUS, VBN and fluoroscopy-guided biopsy with either a 3 mm UTB or a 4 mm TB. In the TB group, small forceps with GS or standard forceps without GS were allowed. Overall diagnostic yield was higher in the UTB group (70.1% vs. 58.7%) and had a shorter procedure duration (24.8 vs. 26.8 min) with fewer complications (2.8% vs. 4.5%) [28]. Again, we observe that multimodal bronchoscopy with the aid of UTB allows for higher diagnostic yield than using TB alone.

**Figure 1 diagnostics-11-02304-f001:**
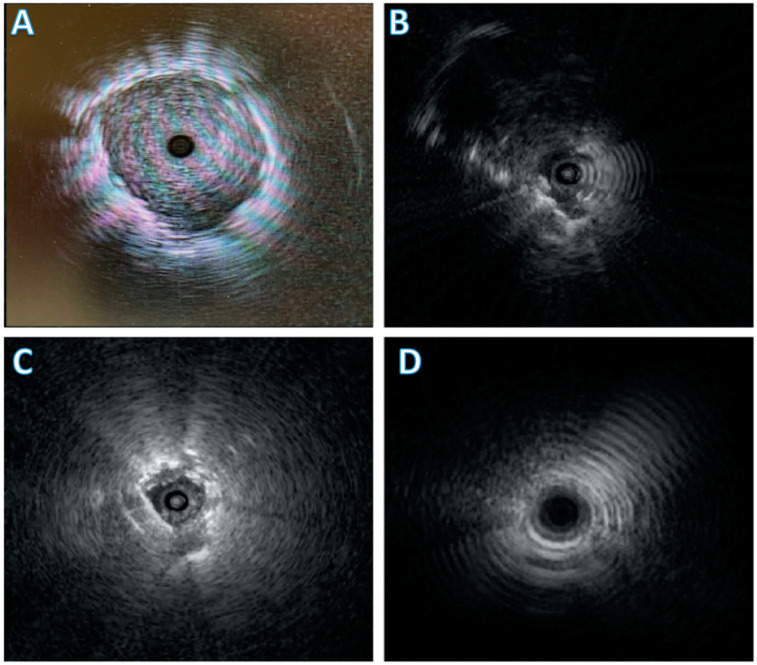
Different views obtainable with radial endobronchial ultrasound (REBUS). (**A**) demonstrates a concentric view with the REBUS probe in the center of the nodule. (**B**) is an eccentric view with the nodule visible off to the side of the REBUS. (**C**) demonstrates a blizzard pattern which can be seen with ground glass nodules [29]. (**D**) demonstrates no view.

## 4. Radial Probe Endobronchial Ultrasound

Radial Probe Endobronchial Ultrasound (REBUS) is a thin, flexible catheter harnessing a rotating ultrasound transducer that produces a 360-degree (“radial”) image that easily passes through the working channel of a therapeutic scope. This allows for real-time localization of lesions that are distal to the physical reach of the bronchoscope. This modality allows for more accurate targeting of a peripheral nodule, reaching it at a distance and providing a 360-degree view in a 2D plane radiating laterally outward from the probe tip [28]. There are typically four different views described in the literature including eccentric and concentric views (described above), ground-glass opacities that may be detected as a “blizzard” pattern [29] and the final view is “no view” or the absence of the nodule. Figure 1 depicts these REBUS views.

A 2011 meta-analysis of REBUS-guided bronchoscopy with 1420 patients reported a pooled diagnostic sensitivity of 73%. Complication rates were similar to non-guided bronchoscopy, with a pneumothorax rate of 1%, with less than half of those requiring chest tube placement [30]. Limitations of studies involved include inconsistencies in patient populations related to cancer incidence and significant inter-operator variability, experience and techniques utilized.

A more recent (2014) single-center, 5-year institutional experience looking at REBUS for peripheral pulmonary lesions found an overall diagnostic yield of 69% for all lesions combined. Yield was directly related to the nodule size, with nodules 2.1–3 cm providing 77% yield and nodules 3.1–4 cm providing 87% yield. When the radial probe provided a concentric view, the diagnostic yield was 84% compared with 48% when the probe had an eccentric view. The most common complication was pneumothorax which occurred in 2.8% of patients—approximately one-half of those required chest tube drainage. Bleeding (>300 cc) occurred in 0.4% of patients and required no additional intervention [31].

The largest meta-analysis assessing REBUS for diagnosis of peripheral pulmonary lesions to date was in 2017 and found a similar overall weighted diagnostic yield of 70.6%. Again, we see that yield was higher in nodules >2 cm, malignant nodules, and those with a positive bronchus sign. Not surprisingly, the yield was higher when the probe had a concentric view rather than eccentric [32].

The benefit of REBUS lies in its ability to provide guided imaging to distal locations, allowing for real-time operator feedback regarding nodule location before the biopsy. Larger nodules and the ability to obtain a concentric view further increases the likelihood for higher diagnostic yield. A major limitation for REBUS is that an eccentric signature tells the operator that the nodule is next to the airway but provides little detail on where the nodule is in 3D space in relation to the eccentric airway in which the REBUS probe is located. Thus, improved techniques such as bronchoscope manipulation and guide sheaths are needed for improved diagnostic yield [33]. In fact, a multicenter randomized study found that diagnostic yield for small, peripheral lesions with REBUS using a GS was higher (55%) than without use of GS (46%) [34].

Another limitation of REBUS is that previous biopsies resulting in focal hemorrhage, atelectasis, or areas into which saline was previously administered can be mistaken for a nodule and lead to false-positive REBUS signal, presenting as a diagnostic drop-off [35]. There is also the fundamental problem of the catheter to nodule deflection (See ENB section for further discussion) when the REBUS is removed from the working channel for biopsy tool advancement.

REBUS allows for more accurate localization of the lesion and positioning of the biopsy instruments by allowing the operator to visualize a nodule undetectable with other modalities. There are many pitfalls to using REBUS, such as the absence of air bronchograms and the potential for misinterpretation of radial ultrasound signals in inexperienced operators.

## 5. Virtual Bronchoscopic Navigation

Virtual bronchoscopic navigation (VBN) is a tool that allows a physician to visualize the anatomy that must be traversed to place them close to the nodule. This procedure relies upon 3D images generated from a helical CT to generate a “roadmap”. Once the “roadmap” is complete, the operator performs bronchoscopy, and the navigational system recognizes the live view of the bronchial tree and synchronizes the virtual and live views. The operator can then follow a pre-determined pathway to the lesion where a biopsy can be performed. When used in combination with ultrathin bronchoscopy, this technique can potentially allow the operator to biopsy more peripheral lesions under direct visualization.

One randomized controlled trial compared VBN-assisted bronchoscopy to non-VBN-assisted bronchoscopy in the ability to diagnose small (<30 mm) peripheral lesions. In the comparison group, the operator used axial CT imaging and site-verification via X-ray fluoroscopy. Although this study did not show a statistically significant difference in the overall diagnostic yield from the VBN-assisted bronchoscopy compared to non-VBN-assisted bronchoscopy (67.1% and 59.9% diagnostic yield, respectively), there was a statistically significant difference in subgroup analysis. When the lesion is in certain areas (right upper lobe or peripheral third of lung field) or when the lesion was complicated to locate radiographically, the VBN-assisted group was superior to the non-VBN-assisted group. Lesions in the RUL showed superior diagnostic yield at 81.3% vs. 53.2% (*p* = 0.004). Lesions in the peripheral third of the lung field had superior diagnostic yield at 64.7% vs. 52.1% (*p* = 0.047). Finally, lesions invisible on posterior-anterior radiographs showed superior diagnostic yield in the VBN-assisted group at 81.3% vs. 53.2% (*p* = 0.004) [36]. This study highlights the utility of VBN-assisted bronchoscopy and the subset of patients in whom it may be beneficial. Another RCT assigned 199 patients to undergo either VBN-assisted or non-VBN-assisted biopsies. The VBN-assisted group had a significantly higher diagnostic yield than the non-VBN-assisted group (80% vs. 67% *p* = 0.03) [37]. One systematic review of VBN in 2014 evaluated the diagnostic yield in all nodules and found an average of 74% as well as an impressive 67% diagnostic yield for smaller nodules (size < 20 mm) [38].

VBN is a useful tool in improving an operator’s ability to accurately and precisely localize small, peripheral lesions. That said, it is not without its drawbacks. Asano et al. performed a non-inferiority trial in 2017 to determine if VBN could be a substitute for X-ray fluoroscopy in the biopsy of peripheral nodules >3 cm in size. The study consisted of 140 patients, and the non-inferiority set point was at 15%. In these individuals, the diagnostic yield was 76.9% in the group utilizing VBN alone, compared to 85.9% in the group that utilized X-ray fluoroscopy to confirm device location. This difference in diagnostic yield of 9% with a 95% CI of −22.3–+4.3% did not meet the non-inferiority set point. The visualization of the nodule via EBUS in the VBN-assisted group was high at 95.4%—similar to the 96.9% noted in the group assisted by X-ray fluoroscopy. However, fluoroscopy was necessary to improve the accuracy during sample collection [39].

The 3D image from the helical CT is the roadmap by which the system guides the operator. There is a significant challenge with VBN if there is a discrepancy between the CT images and the real-time bronchoscopic images as the system has no way of updating or correcting in real time [40]. This concept of CT-to-body divergence is a recurrent theme for many of our methods of guided bronchoscopy. VBN helps the operator to localize their tools in the bronchial tree, but outside cases of a specific subset of peripheral nodules, it falls short of the idealized biopsy tool [41]. There is no tool-tip locator by which the operator may find their way back to the “road”. That technology, however, does exist. Electromagnetic Navigational Bronchoscopy (ENB) is the “GPS” update to the “map” that is VBN.

## 6. Electromagnetic Navigational Bronchoscopy

In this system, similar to VBN, a pre-procedure CT scan of the chest is performed to produce a virtual trachea-bronchial tree. The operator identifies the target lesion then plots a pathway to the lesion via nearby airways. What differentiates ENB from VBN is that an electromagnetic probe at the tip of a guide sheath/navigation catheter is used to track progress toward the target, rather than computerized recognition of progress according to visual appearance of the airways utilized by VBN. This electromagnetic field is generated either by a board on which the patient lies or by sensors placed on and around the patient [42]. Currently, there are two commercially available ENB systems: SuperDimension (Medtronic, Minneapolis, MN, USA) and SPiNDrive (Veran Medical Technologies, Inc., St. Louis, MO, USA).

In the SuperDimension system, a locatable guide (LG) is affixed to the distal aspect of a guide sheath with curved tip called an extended working channel (EWC), which is then advanced via a therapeutic bronchoscope. This system then guides the operator to the lesion, where the guide sheath remains in place and the LG is removed. The operator may then biopsy at this location; however, they will not have further guidance from the system once the LG is removed. Because the LG is stiff and tends to straighten the EWC, the operator risks missing the lesion due to catheter-to-nodule deflection as the EWC channel curve returns [43]. Newer models attempt to solve this problem by installing guide-chips on the biopsy tools and/or the navigation catheter itself [44].

In the NAVIGATE trial, 1215 subjects with peripherally located pulmonary nodules were enrolled in 29 academic and community hospitals across the United States to determine the safety and efficacy of ENB. The median nodule size was 20 mm, and among 1157 who underwent an ENB-guided biopsy, 94% had tissue obtained, 99% of patients completed their 1-month follow-up, and 80% completed the 12-month follow-up. The 12-month diagnostic yield was reported at 73%. Sensitivity, specificity, positive predictive value, and negative predictive value for malignancy were reported at 69%, 100%, 100%, and 56%, respectively. These procedures were tolerated well with low rates of complications and provided the diagnosis in nearly 75% of cases [45].

In addition to catheter-to-nodule deflection, there is often a discrepancy between the inspiratory and expiratory phase-CT scans. That discrepancy has shown the “movement” of pulmonary nodules up to 2.5 cm when located in the lower lobes [43]. Another hurdle for the successful diagnosis of peripherally located pulmonary nodules is atelectasis [46]. This simple problem is one of the main reasons it is essential for the operator performing this biopsy to be experienced in this type of procedure. Expert strategies for optimizing the ventilator to reduce the risk of atelectasis, as well as operator speed in reaching and performing the biopsy, may be the difference between successful tissue acquisition and the need for further procedures. Even when put in the hands of an experienced operator and anesthesia team working to minimize atelectasis, ENB and VBN still rely on the pre-operative CT rather than real-time imaging. ENB may use 2D fluoroscopy during the procedure, but that imaging is often insufficient to visualize these small and peripheral lesions. To aid in that real-time visualization during the procedures, some operators reach for digital tomosynthesis.

## 7. Digital Tomosynthesis

Digital tomosynthesis—also known as fluoroscopic navigation—helps identify the lesion in real time while also providing feedback on the instrument’s location and distance to the lesion. The 3D representation recreated during this procedure comprises a series of images obtained while a c-arm rotates a specified number of degrees around a patient (usually between 20–60 degrees). Although this sounds similar to a typical multi-slice CT scan, it is different in a few important ways. The rotation angle, number of images, and radiation exposure allow for an accurate and precise representation of the lesion in real time. The effective radiation dose is approximately 0.1 mSv which is about double that of a traditional 2-view CXR. However, it is about 1/10th–1/40th the dose of a typical chest CT [47]. In 2010, 228 patients were enrolled in a study to compare the diagnostic utility of digital tomosynthesis compared with radiography and CT was used as the reference standard. Accuracy differed in this study in radiography vs. digital tomosynthesis by 43% vs. 90% and 49% vs. 92%, respectively, for the two radiography readers who participated in the study [48]. This technology was further tested in the analysis of 168 nodules at one medical center. This was a before-and-after study, where 101 cases were performed using ENB and 2D-fluoroscopy, and then digital tomosynthesis was implemented for the remaining 67 cases. This study has the advantage of being performed in the same medical center, by the same operators with a similar patient demographic, thus reducing many of the confounding variables seen in other studies. The nodules were similarly located, and the size of the nodules was similar as well (median 15 mm vs. 16 mm). The diagnostic yield using traditional ENB with 2D-fluoroscopy was 54%, and after utilizing digital tomosynthesis, the diagnostic yield improved to 79% (*p* = 0.0019). Complication rates during this study were low, with only 1.5% of cases resulting in a pneumothorax and the between-group difference was negligible [49].

A retrospective study of 324 patients who underwent digital tomosynthesis-assisted ENB for indeterminate lung nodules found a diagnostic yield of 82%. The average nodule size was 1.9 cm, and the majority (65%) were in the peripheral third of the lung. A bronchus sign was present in only 24% of cases. In a more conservative analysis considering all nodules lost to follow-up as false negatives, the diagnostic yield remained high at 77%. Complications included eight pneumothoraces (2.5%) and one episode of respiratory failure [50]. This study is important in that it provides a more conservative yield representation by labeling biopsies as non-lesional if they are, for instance, normal lung tissue or mild inflammation or were initially negative but lost to follow-up. When applied to other studies, this method of calculating diagnostic yield can reveal significant disparities in how authors determine their yield, specifically when follow-up is used to confirm benign histology.

## 8. Cone-Beam Computed Tomography

As these incredible advancements are brought into the medical world, sometimes the biggest hurdle is finding a way to apply them efficiently and safely. CT scans offer incredible anatomic detail of the pulmonary system, but due to the size and shape of the machine, it can be difficult for the operator to work around these machines. If an operator were to use a traditional fan-beam CT to guide their tools to a lesion, the patient could be exposed to dangerously high radiation levels with repeat scans. Cone-beam CT solves this problem by using a flexible C-arm, which is lower power and utilizes a flat-panel detector. CBCT scans images in a cone shape rather than the fan shape of traditional CT [51]. This allows for a narrower rotation (usually 180 degrees) than the traditional CT scans (360-degrees), and with that 200–300 times less radiation. These images are then reconstructed to provide a three-dimensional image [52]. This technology is becoming more prevalent as the advantages are realized.

One significant advantage to CBCT is that it reduces the risk of CT-to-body divergence since the CT scan is carried out intra-procedurally. As the many contributors to nodule motion occur during the procedure (respiro-phasic changes, atelectasis, and regional collapse of target lobe due to “over-wedging”), the operator can reimage, adjust if necessary, and resample the targeted area [40]. Additionally, CBCT offers a bypass to instrument deflection by allowing for real-time acquisition of tool in lesion images (Figure 2).

While CBCT allows for excellent real-time imaging capture, the use of guided (augmented) fluoroscopy may offer a higher diagnostic yield and safety profile. Essentially, the patients undergo CBCT imaging, and 3-dimensional segmentation of the nodule is performed using proprietary software. Then, during ENB, the lesion is projected as an overlay on live 2D fluoroscopy, allowing for real-time image guidance. In the largest retrospective study to date involving CBCT guided bronchoscopic biopsy by Pritchett et al. of 93 lesions, the overall DY was 83% using this technique. There was no independent correlation between DY and lesion size, location, visibility under standard fluoroscopy and presence of bronchus sign. The pneumothorax rate was 4% [53]. This study affirmed that augmented fluoroscopy with CBCT imaging is safe and allows for high DY during ENB guided biopsy.

Another study used a combination of CBCT, ENB, REBUS with or without a transbronchial access tool to biopsy lesions with a similar yield of 77.2% [54]. One of the most impressive yields observed was 90% when utilizing CBCT guided ultrathin bronchoscope with VBN. The biopsy tool was within or adjacent to the target lesion under CBCT guidance in 95% of cases [55].

A smaller study of 26 patients looked at “needle in lesion” as a primary outcome to determine the accuracy of F-ENB as confirmed by CBCT. When ENB was performed along with digital tomosynthesis (F-ENB) followed by CBCT once the needle was in the expected location, this resulted in a “needle in lesion” in 72% of cases. Mean nodule size was 13 mm, the majority were in the peripheral third of the chest (83%), and 17% had a bronchus sign. There were no complications. While this was a smaller study, it further elucidated that CBCT can help confirm the needle is intralesional and that F-ENB has high accuracy, even in cases of small peripheral nodules [56].

**Figure 2 diagnostics-11-02304-f002:**
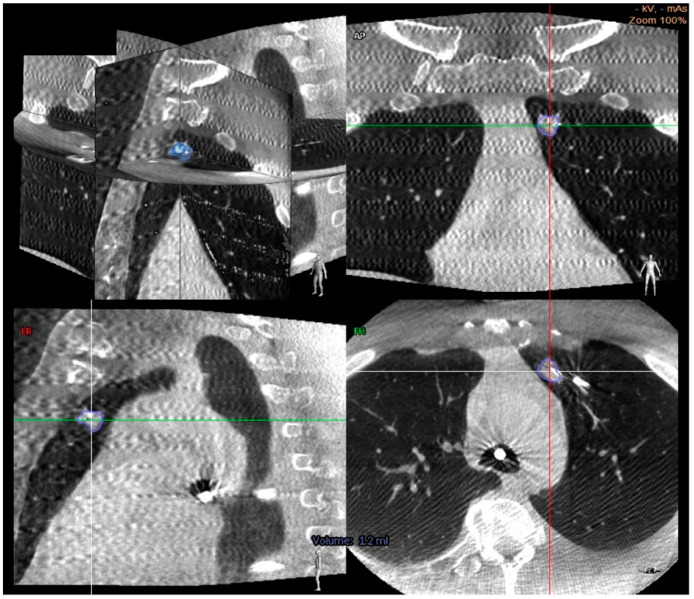
Cone beam CT images demonstrating a center strike represented by a needle within the nodule in axial, coronal, and sagittal planes [56].

## 9. Robotic Bronchoscopy

Robotic-assisted bronchoscopy was first introduced to the world of interventional bronchoscopy via the Monarch system in 2018, and we already see more examples entering the market, such as the ION Endoluminal System. The Monarch endoscopic system attempts to increase diagnostic yield by addressing the specific areas where bronchoscopy has fallen short in years past. Their system can reach deeper segments of the bronchial tree than traditional bronchoscopy by 4.2 cm. Additionally, their system provides the operator with greater control in these distal segments allowing them to make turns and enter airways previously unreachable by traditional bronchoscopy. This system has traditional white-light vision abilities, and with that, it also has suction and irrigation to improve visibility. Additionally, it uses electromagnetic navigation, pattern recognition software, and robotic kinematic data to pinpoint the tip location in a patient’s bronchial system and guide the operator to the lesion [57].

In 2019, this system was tested in eight cadavers with artificial tumor targets ranging in size from 10–30 mm. The purpose of this study was to evaluate the robotic system’s ability to reach these small, peripheral nodules. Sixty-seven artificial tumors were identified and pursued, and the same operator attempted each procedure. They discovered that robotic bronchoscopy—combined with EBUS, REBUS, and fluoroscopy—achieved lesional biopsies in 97% (65/67) of the identified nodules. It is important to note just how significant that 97% diagnostic yield is compared to previously reported data just two years prior of between 30–40% [58].

A recent multicenter pilot and feasibility study looked at robotic bronchoscopy for biopsy of peripheral lesions (size 1–5 cm) and used successful lesion localization with REBUS as the primary endpoint rather than diagnostic yield. Lesion localization was successful in 96.2% of cases with pneumothorax complication rate of 3.7% [59]. While this study proved the safety and feasibility of lesion localization, it did not touch on the diagnostic yield. One such study that did look at yield as a primary endpoint was out of the Veterans Affairs Health System in Pittsburg. This was a retrospective study of 25 patients that underwent robot assisted bronchoscopy with biopsy and found a diagnostic yield of 96% with no complications [60]. It is important to note that in this study, normal lung histology was not excluded from yield calculation.

Each device introduced thus far has taken advantage of the technologies that preceded it and built upon them to create a more advanced and sensitive machine. The majority of all navigational procedures include aspects of each intervention we have discussed thus far. It is not uncommon for an operator to begin a procedure with flexible fiberoptic bronchoscopy for airway inspection, move on to EBUS for mediastinal and centrally located lesions, and then move on to other advanced modalities to reach small peripheral nodules. Each technology attempts to provide a more accurate nodule image, correct for CT-to-body divergence and respiratory variability, all while minimizing radiation exposure.

Another example of novel bronchoscopic technology is shape sensing, designed by Philips. Shape sensing solves the problems of CT-to-body divergence and respiratory variability by entirely changing how we identify locations. The tools used during this procedure are embedded with fiberoptic wires [61]. A pre-procedure CT scan is still required to identify the operator-identified target-lesion and pathway, but from that point forward, the operator is guided by the system. It recognizes its location in 3D space via variations in light intensity traveling through the fiberoptic cables, which correlates to a specific shapes and bends in the catheter as it moves through the bronchial tree. That shape correlates with a location, and in early studies, that location has proven to be accurate to under a millimeter [62]. The PRECISION-1 study published in 2020 was a prospective, single-blinded, randomized controlled trial comparing different methods of sampling peripheral pulmonary nodules. The study compared robotic bronchoscopy vs. electromagnetic navigational bronchoscopy vs. ultrathin bronchoscopy with REBUS. Cadaveric models with 20 nodules measuring an average of 1.6 cm in diameter and located throughout all five lobes (80% in the periphery) were sampled using one of these three techniques. Sixty procedures were performed with the primary endpoint of successful localization and sampling of the target nodule. The secondary endpoint was the distance from the target nodule. The successful localization and sampling rate was superior in Robotic bronchoscopy to ENB (80% vs. 45%, *p* = 0.02). Ultrathin bronchoscope with REBUS had a significantly lower yield at only 25% [63].

In a safety and feasibility study, the ION robotic bronchoscopy system with shape-sensing technology was first tested in humans in 2019. In this study, 29 subjects with peripheral lesions of a mean size of 1.2 cm underwent sampling. Even though 41.4% of these cases lacked CT bronchus sign, the target lesion was reached in 96.6% of cases. Furthermore, a diagnostic yield for malignancy was noted in 88% of cases [64]. This system safely navigated to the peripheral lesion through enhanced dexterity, positional awareness, and stability, allowing the operator to achieve a high diagnostic yield.

## 10. Bronchoscopic Transparenchymal Nodule Access

Despite these imaging and guidance advancements, there are occasions where an airway simply does not lead to a pulmonary nodule. Bronchoscopic transparenchymal nodule access (BTNA) attempts to solve this problem by creating a pathway between an airway and a lesion. The system utilizes a CT scan to create a path for the operator to guide the bronchoscope, not unlike some other methods of navigational bronchoscopy. Rather than guiding the operator directly to the lesion, it guides to a location—the point of entry (POE). A needle is then used to gain access through the bronchus into adjacent lung parenchyma, and a balloon dilates the space to allow a sheath to pass. The system attempts to find the most direct path while also avoiding vasculature [65]. In the first human study performed in 2014, the diagnostic yield was 83%, and there were no pneumothoraxes or significant bleeding reported [66]. The University of Heidelberg performed a single-arm intervention, prospective study two years later to determine if BTNA could be utilized in patients with small (<30 mm) peripheral nodules located in areas previously inaccessible. Six patients were identified and recruited, and in five of them, a tract was created between the POE identified by the system (LungPoint) and the nodule. The tract length ranged from 11 to 46 mm, and a successful biopsy was obtained in all five patients in whom a tract was created [65]. Two patients had pneumothoraces after the procedure, one of which required intervention (40% pneumothorax).

## 11. Discussion

Interventional Pulmonology is an important growing field and is rapidly being recognized for the value it adds to patient care. New technologies are being introduced faster than they can be studied rigorously. Determining which technology is best remains a challenge with the available literature. We can see diagnostic yield vary greatly and case series which are confounded with cases selection bias. This is well illustrated by the superior performance of sputum cytology compared to much more invasive modalities. It is worth mentioning that there are no standard guidelines in the bronchoscopic literature for reporting and calculating diagnostic yield, making comparisons across sources very challenging. Furthermore, even when two studies use similar statistical analysis for nodule reporting and bronchoscopic techniques, they may vary in the biopsy technique utilized (for example forceps size). This poses yet another limitation for comparison between studies and is a significant confounder.

Early in this article, we described the need for a diagnostic tool that can safely differentiate benign from malignant lung nodules, provide staging, and, when needed, enough material for molecular testing. Currently, we have two primary minimally invasive modalities of sampling peripherally located, small pulmonary nodules, transthoracic needle aspiration and bronchoscopic biopsies. The advanced technologies discussed above have all been developed to improve the diagnostic yield for lung nodules so lung cancer can be rapidly and quickly diagnosed. There are many hurdles in achieving this goal while maintaining the procedure’s minimally invasive nature, such as minimizing radiation exposure, overcoming CT-to-body divergence, catheter deflection, and respiratory variability. Many of these technologies sound appealing in approaching those problems, and it is easy to falsely ascribe superiority to one or the other based on their claims. However, in the absence of large-scale, prospective randomized trials to compare diagnostic yield reliably, it is impossible to declare a “best method”. Performing such a study for the technologies is challenging as they are frequently used together and constantly evolving, with technologies and techniques arriving faster than rigorous studies can be performed. Another challenge is the significant heterogeneity between studies in how diagnostic yield is calculated and reported. For example, Table 1 lists various studies which often differ significantly in what histology was reported, their follow up duration and whether normal lung histology was included in yield calculation. A mutually agreed-upon standard for calculating and reporting diagnostic yield would be of great value to allow for some comparison across the literature.

To illustrate the value of a standard reporting system, we present the following example in a hypothetical case series of 100 nodules with a cancer prevalence of 10% (n = 10). All 100 nodules could be missed, and after interval follow-up, 90% would be confirmed to be benign by radiographic stability or resolution. This can be reported as a diagnostic accuracy of 90% despite no nodules being accurately sampled.

We would propose that all histology be evaluated after the biopsy, and if the histological findings could not reasonably represent the radiographic appearance of a nodule (“lesional histology”), the biopsy is treated as a false negative and cannot be counted as a true negative regardless of future radiographic behavior. When this standard is applied to prior studies, we can see a dramatic drop in their yield. Additionally, cases with benign lesional histology can later be determined to be malignant. For example, one study looked at long-term follow-up of CT-guided TTNB specimens and found that 51% of all “negative” biopsies were, in fact, false-negative for malignant diagnosis [66]. A standard time interval of follow-up of these nodules prior to reporting yield is additionally invaluable in making an accurate assessment of the diagnostic yield.

## 12. Conclusions

A standard method for determining diagnostic yield in biopsy literature would be valuable in guiding future practices to ensure we provide the best possible care for our patients. The diagnostic yield and safety profile are the essential determinants in what we should ultimately recommend to our patients. The status of the current literature does not provide significant evidence to make a strong recommendation for our patient. Currently, local expertise and shared decision-making are frequently utilized to guide management. The major hurdles in recommending evidenced based “best practices” guidelines for biopsy of a peripheral nodule lies in the inherent variability between studies. Bronchoscopic technologies are frequently used in tandem and biopsy techniques inevitably vary between operators. Additionally, there is significant variability in how diagnostic yield is calculated and reported. Large scale, prospective and randomized studies that utilize an agreed upon standard for calculating and reporting diagnostic yield would be of high clinical value and allow providers to achieve a more accurate understanding of which technologies may be superior in their safety profile and offering the highest diagnostic yield.

## Figures and Tables

**Table 1 diagnostics-11-02304-t001:** A list of studies examining the diagnostic yield of various bronchoscopic techniques for the diagnosis of lung nodules. Note the heterogeneity in defining adequate follow up duration and what is considered diagnostic histology.

Author Year	Technologies Investigates	Number of Nodules	Diagnostic Yield	Follow Up Duration	All Histology Reported?	Normal Lung Histology Excluded from Yield Calculations
Folch et al. [45]	EMN	1344	73%	12 months	Yes	No
Asano et al. [36]	VBN w/ultrathin	26	65%	16 months	Yes	Yes
Katsis et al. [50]	Digital Tomosynthesis-assisted NB	363	82%	12 months	Yes	Yes
Chen et al. [31]	REBUS	438	69%	60 months	Yes	No
Asano et al. [39]	VBN w/XRF	64	85.9%	24 months	No	No
Pritchett [53]	Cone beam CT w/ENB	93	83.7%	9 months	Yes	Yes
Oki et al. [27]	UTB w/VBN	150	74%	33 months	Yes	Yes
Oki et al. [34]	EBUS, +/− GS	300	55.3%	12 months	Yes	Yes
Chen et al. [59]	Robot	54	74.1%	12 months	Yes	Yes
Ekeke et al. [60]	Robot	25	96%	6 months	Yes	No

Legend: EMN = electromagnetic navigation; VBN = virtual bronchoscopic navigation; REBUS = radial probe endobronchial ultrasound; XRF = x ray fleurosocopy; UTB = ultrathin bronchoscopy; GS = guide sheath.

## Data Availability

Not applicable.

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
