# Peer review of "Advanced Bronchoscopic Technologies for Biopsy of the Pulmonary Nodule: A 2021 Review"

_diagnostics, 2021, doi:10.3390/diagnostics11122304_

Round 1

Reviewer 1 Report

I want to thank the handling editor for providing me the opportunity to review for this
distinguished journal. I would also like to congratulate the authors for their elegant manuscript
and comprehensive review.

This is a narrative review of bronchoscopic technologies related to the diagnosis of pulmonary
nodules, especially those developed in the last decade. The authors cover all bronchospic
technologies, including thin and ultra-thin bronchoscopy, radial probe endobronchial
ultrasound, virtual bronchoscopic navigation, electromagnetic navigational bronchoscopy,
robotic bronchoscopy, and bronchoscopic transparenchymal nodule access.

The topic is of high importance not only to scholars and clinicians but also to a general
audience, since it investigates diagnostics of a common radiologic entity, namely lung nodules.

Many of the authors of the paper appear to be senior researchers, after a cursory review of their
published articles, from US institutions with a plethora of publications in the field of
interventional pulmonology.

The title of the manuscript describes the investigated topic adequately and accurately.

The authors discuss the topic in an unbiased manner, covering its most significant aspects. A
more in-depth analysis of virtual bronchoscopic navigation and electromagnetic navigational
bronchoscopy would be possible; however, this could result in a lengthy article that would
hinder reading. The adequate coverage of the topic is reflected by the 79 references, with most
of them dated within the last 10 years. In addition, the majority of the papers cited are from
journals with high impact in their respective field, such as the New England Journal of
Medicine, Chest, and the Annals of Translational Medicine.

The conclusions drawn are a logical sequence of the content and identify areas for future
research.

Finally, I have no concerns regarding adherence to ethical guidelines. One of the authors has
potential conflicts of interest that are clearly stated for the reader to be aware of.

Overall, this is a high-quality review on an important topic that can attract readings and
citations for your journal. I recommend its acceptance in its current form.

Author Response

Thank you for your comments. Your reviews help us write a better, more impactful manuscript and we genially appreciate your time and feedback.

Reviewer 2 Report

  1. your systematic review has innapropiate auto citation - 3 articles from James Katsis, including one in the table 1 - article no 77.
  2. the abstract does not respect the IMRAD structure. Also, the entire article is not structured according to PRISMA CRITERIA. 
  3. You used some abreviations that are not explained in the text the first time used- for example, CT in the 41 line, EMN in the 146 line, RUL in the 265 line, TBAT in the 395 line. In addition, you should specify the explanation for some abreviations first time it is use in the article - for example TB and NTB in the 169-170 were used before. 
  4. There are some paragraphs without bibliography - for example lines 81-87, 138-142, 174-177, 325, 363-365, 374-377, etc.
  5. in the line 508 - there are two numbers - 74,75 - i didn't understand what is their significance. 
  6. you didnt'n mention the bibliography for the figure 1 and 2. 
  7. taking into account the subject, you included in the table 1 too few studies, from whom one is a meta analysis and one is autocitation. You can not include in a systematic review other systematic reviews or meta analysis, only clinical trials or case reports.
  8. the aim of the article is not well defined. also, the conclusion does not reply to the aim - both should be reformulated; the aim of the review should be mentioned at the end of the introduction, so the information from the 44-49 lines should be at 109-115 lines. 
  9.  regarding the terms of "eccentric/concentric" you mentioned it at page 4 but explained it in the page 5.. In my opinion, you should explained it before it was first time used. 
  10. The studies 77,78,79 are only mentioned in the table 1 but are not discussed in the results or discussion section.

Author Response

Thank you for your comments.  Your reviews help us write a better, more impactful manuscript and we genially appreciate your time and feedback.  Please see our response to your comments below. 

Reviewer Comments:

  1. your systematic review has inappropriate auto citation - 3 articles from James Katsis, including one in the table 1 - article no 77.
  • We are unsure how to resolve this issue. We feel these papers are important to the review as they include the largest case series of F-NAv to date.  If needed Dr. Katsis can recuse himself as a co-author for this review. 

  1. the abstract does not respect the IMRAD structure. Also, the entire article is not structured according to PRISMA CRITERIA. 

This publication is not a systematic review but rather an introduction to the some of the nex technogies that were introduced to the field.  We did not intent to make a direct comparision across trials and think it would be inappropriate to do so given the significant heterogeneity in how diagnostic yield is reported.  We will make this more clear in the abstract. 

  1. You used some abreviations that are not explained in the text the first time used- for example, CT in the 41 line, EMN in the 146 line, RUL in the 265 line, TBAT in the 395 line. In addition, you should specify the explanation for some abreviations first time it is use in the article - for example TB and NTB in the 169-170 were used before. 

Thank you, we have added explanations for these abbreviations and fixed the redundant abbreviations. 

  1. There are some paragraphs without bibliography - for example lines 81-87, 138-142, 174-177, 325, 363-365, 374-377, etc.

Citations were added for the 81-87.

138-142 was edited down and. Included in the prior. Paragraph with citations. 

174-177 has a ciation to 25 - Nishii Y, Nakamura Y, Fujiwara K, et al. Use of Ultrathin Bronchoscope on a Need Basis Improves Diagnostic Yield of Difficult-to-Approach Pulmonary Lesions. Frontiers in Medicine. 2020;7(December):1-7.

325 – ciations to #41,42

363-365 – ciations 45-47

374-377 - #47

  1. in the line 508 - there are two numbers - 74,75 - i didn't understand what is their significance. 

Fixed

  1. you didnt'n mention the bibliography for the figure 1 and 2. 

Figures 1 and 2 are both the creations of james katsis.  Figure 1 has been edited for use in this paper. 

We have added the citation for the “blizzard pattern”. From Blizzard sign, you may want to cite the article by Izumo, et al. (European Respiratory Journal 2015; 45: 1661-1668).

  1. taking into account the subject, you included in the table 1 too few studies, from whom one is a meta analysis and one is autocitation. You can not include in a systematic review other systematic reviews or meta analysis, only clinical trials or case reports.

Our goal is not to do perform a meta or systematic analysis as we feel they would both be significantly flawed as a result of poor quality studies with varying methods of determining diagnostic yield.  We aimed to exhibit this issue in the literature and. Hope that a consensus methodology would be developed to ensure technological comparisons could potentially be done in the future. 

Meta analysis were reviewed from the table

  1. the aim of the article is not well defined. also, the conclusion does not reply to the aim - both should be reformulated; the aim of the review should be mentioned at the end of the introduction, so the information from the 44-49 lines should be at 109-115 lines. 

Additional information was added to the Introduction to clarify our objective. 

  1.  regarding the terms of "eccentric/concentric" you mentioned it at page 4 but explained it in the page 5.. In my opinion, you should explained it before it was first time used. 

The redundant explanation was removed.

  1. The studies 77,78,79 are only mentioned in the table 1 but are not discussed in the results or discussion section.

These will be incorporated into their respective sections. 

Reviewer 3 Report

The authors have made a comprehensive and detailed report on the development and new techniques of bronchoscopic biopsy.

Comments:

  1. Assign the A~D to pictures of Figure 1.
  2. For the Blizzard sign, you may want to cite the article by Izumo, et al. (European Respiratory Journal 2015; 45: 1661-1668).
  3. P6, L231: Recently, the results of a randomized controlled trial of guide sheath (GS) vs. non-GS was published, showing the superiority of GS (Oki M, et al. Guide sheath versus nonguide sheath method for endobronchial ultrasound-guided biopsy of peripheral pulmonary lesions: A multicenter randomized trial. Eur Respir J. 2021 Oct 8:2101678). Please consider citing this as the most recent reference.

Author Response

Thank you for your comments.  Your reviews help us write a better, more impactful manuscript and we genially appreciate your time and feedback.  Please see our response to your comments below. 

Reviewer Comments:

  1. Assign the A~D to pictures of Figure 1.

Completed, thank you

  1. For the Blizzard sign, you may want to cite the article by Izumo, et al. (European Respiratory Journal 2015; 45: 1661-1668).

Added citation, thank you.

  1. P6, L231: Recently, the results of a randomized controlled trial of guide sheath (GS) vs. non-GS was published, showing the superiority of GS (Oki M, et al. Guide sheath versus nonguide sheath method for endobronchial ultrasound-guided biopsy of peripheral pulmonary lesions: A multicenter randomized trial. Eur Respir J. 2021 Oct 8:2101678). Please consider citing this as the most recent reference.

We will add this reference.  Thank you.

Reviewer 4 Report

Comments and suggestions

Dear Editors,

I would like to thank you for the opportunity to evaluate the review of the colleagues Levine M. et al . Enclosed you will find my comments:

In principle, the authors have described the topic of different bronchoscopic technologies, in particular the emerging new procedures well and evaluated them in an adequate neutral way.

Minor comments:

So far, many citations are duplicated or outright incorrect, so a comprehensive correction and reevaluation of the individual citations needs to be done. In addition, the citations should be listed in sequential order, this has not been done stringently so far.

Line 37: Reference 4 is incorrect

Line 57: Please cite both studies mentioned here

Line 65: Insert reference to the Nelson- Trial

Reference 7 and 8 are identical

Reference 14 does not appear in the text

Reference 9 and 14 are identical

Line 137: Reference 19 is incorrect, reference 23 is referred to here.

Line 237: Reference 39 is wrong

Line 276: Reference incorrect

Line 363: Check reference 59

Line 508: Please superscript reference numbers

Furthermore, care should be taken to ensure that all abbreviations are introduced once

Line 113: VBN is introduced, but VN is used in line 145

Line 114: ENB is introduced, but EMN is used in line 146.

Line 265: RUL not introduced

Line 272: RCT not introduced

Line 395: TBAT not introduced

Line 400: F-ENB not introduced

Line 418: DY not introduced

Line 510: BTNA not introduced

Line 530: TTNA not introduced

It is clear that the present review focuses primarily on bronchoscopic techniques. However, since the focus is on the diagnosis of pulmonary nodules, it should be mentioned that the results obtained may not be influenced solely by the bronchoscopic technique used, but also by the biopsy technique (small forceps, large forceps, cryobiopsy, size of TBNA needle,  ....) during bronchoscopy. The studies listed often do not even mention the biopsy technique used, but this may contribute to a relevant bias and should be introduced as a limitation for comparison.

Lines 274-276: The additive use of EBUS should be mentioned.

In the description of electromagnetic navigation, the potential problem of atelectasis is discussed. Here, the authors refer to velitaion associated atelectasis. However, electromagnetic navigation can be performed even in flexible intubation and under conscious sedation with continued spontaneous breathing, thus bypassing the problem of atelectasis formation. Otherwise, atelectasis formation would be much more in focus in any bronchoscopic technique, too, and thus would not be a leading problem in electromagnetic navigation.

Line 439: Legend for Figure 2: Please correct Ct to CT. Please delete the second point at the end of the legend.

Line 462-470: This section listed under “Robotic Bronchoscopy” describes the sequential use of different bronchoscopic techniques. This is not limited to robotic bronchoscopy. Therefore, this should be listed separately, under a separate subsection if necessary, or in the discussion.

There is no reference to Table 1 in the text.

Table 1:

- The sorting of the literature seems confusing and unclear. Please preferentially list the literature according to the sequence in the text

- Shinagawa N et al. is not assigned to any reference

- Chen et al: correct citation

- Please list all abbreviations in the table separately in a legend despite previous mentioning in the text

References:

Please correct incorrect, incomplete or doubled citations as the following and check for the others: Ref.  4, 7=8, 9=14, 19=21, 32, 33, 42, 43, 53, 59, 65, 69, 70, 71, 72, 74, 75.

In addition, references 76 to 79 are not mentioned in the manuscript. Please add them to the text or delete the references.

Thank you for having the opportunity reviewing this manuscript.

Author Response

Thank you for your comments.  Your reviews help us write a better, more impactful manuscript and we genially appreciate your time and feed back.  Please see our response to your comments below. 

Line 37: Reference 4 is incorrect

- fixed

Line 57: Please cite both studies mentioned here

- fixed

Line 65: Insert reference to the Nelson- Trial

- fixed

Reference 7 and 8 are identical

- deleted one of them 

Reference 14 does not appear in the text

- removed

Reference 9 and 14 are identical

- as above

Line 137: Reference 19 is incorrect, reference 23 is referred to here.

- fixed

Line 237: Reference 39 is wrong

-fixed

Line 276: Reference incorrect

-fixed

Line 363: Check reference 59

- updated reference 

Line 508: Please superscript reference numbers

- fixed

Furthermore, care should be taken to ensure that all abbreviations are introduced once

Line 113: VBN is introduced, but VN is used in line 145

- fixed

Line 114: ENB is introduced, but EMN is used in line 146.

- fixed

Line 265: RUL not introduced

- fixed

Line 272: RCT not introduced

- introduction added 

Line 395: TBAT not introduced

- removed abbreviation  

Line 400: F-ENB not introduced

- introduced

Line 418: DY not introduced

- introduced 

Line 510: BTNA not introduced

- introduced

Line 530: TTNA not introduced

- removed abbreviation 

It is clear that the present review focuses primarily on bronchoscopic techniques. However, since the focus is on the diagnosis of pulmonary nodules, it should be mentioned that the results obtained may not be influenced solely by the bronchoscopic technique used, but also by the biopsy technique (small forceps, large forceps, cryobiopsy, size of TBNA needle,  ....) during bronchoscopy. The studies listed often do not even mention the biopsy technique used, but this may contribute to a relevant bias and should be introduced as a limitation for comparison.

- we will add comments about biopsy technique as a potential confounder in comparison. 

In the description of electromagnetic navigation, the potential problem of atelectasis is discussed. Here, the authors refer to velitaion associated atelectasis. However, electromagnetic navigation can be performed even in flexible intubation and under conscious sedation with continued spontaneous breathing, thus bypassing the problem of atelectasis formation. Otherwise, atelectasis formation would be much more in focus in any bronchoscopic technique, too, and thus would not be a leading problem in electromagnetic navigation.

- we will either take out this portion of the paper or expand on it as you payed out. 

Line 439: Legend for Figure 2: Please correct Ct to CT. Please delete the second point at the end of the legend.

- fixed 

Line 462-470: This section listed under “Robotic Bronchoscopy” describes the sequential use of different bronchoscopic techniques. This is not limited to robotic bronchoscopy. Therefore, this should be listed separately, under a separate subsection if necessary, or in the discussion.

- we used it as an introduction to the final topic. We will either be more clear in that it is not limited to robotics alone, or we will move it to the discussion section as is recommended.

There is no reference to Table 1 in the text.

- We will add it in the discussion section 

Table 1:

  • The sorting of the literature seems confusing and unclear. Please preferentially list the literature according to the sequence in the text
  • Shinagawa N et al. is not assigned to any reference - fixed

- Chen et al: correct citation - fixed 

  • Please list all abbreviations in the table separately in a legend despite previous mentioning in the text
    • we will add a legend 

References:

Please correct incorrect, incomplete or doubled citations as the following and check for the others: Ref.  4, 7=8, 9=14, 19=21, 32, 33, 42, 43, 53, 59, 65, 69, 70, 71, 72, 74, 75.

- corrected

In addition, references 76 to 79 are not mentioned in the manuscript. Please add them to the text or delete the references.

  • corrected

Thank you again for your time; we greatly appreciate it. 

Round 2

Reviewer 2 Report

Congratulation for your hard work!